# Exploring Medication Adherence Using M-Health: A Study from Veterinary Medicine

**DOI:** 10.3390/pharmacy8010038

**Published:** 2020-03-10

**Authors:** Marta Ribas, Ana Mafalda Lourenço, Afonso Cavaco

**Affiliations:** 1Faculty of Pharmacy, University of Lisbon, 1649-003 Lisboa, Portugal; mr_martinha12@hotmail.com; 2Faculty of Veterinary Medicine, University of Lisbon, 1300-477 Lisboa, Portugal; anamafalda@fmv.ulisboa.pt

**Keywords:** medication adherence, canine atopic dermatitis, pruritus, veterinary pharmaceuticals, mobile health

## Abstract

**Background:** Pharmacy practice includes the handling of human and animal medication. Amongst veterinary pharmaceutical treatments, the management of Canine Atopic Dermatitis (CAD), a chronic skin condition affecting 10%–15% of the canine population, is complex and demanding. Medication regimens are tailored to each animal and their owner or caregiver. The purpose of this study was to assess the impact of a mobile health (m-health) application (Petable^®^) to support the medication adherence in CAD treatment and clinical improvement. **Methods:** A total of 30 atopic dogs under treatment for CAD and their caregivers were enrolled. Both the dogs’ and owners’ background data were recorded as well as clinical and medication adherence information. This was accomplished by direct observation, clinical files consultation, mobile application, and medication adherence (*Medida de Adesão aos Tratamentos*—MAT questionnaire) feedback. **Results:** The overall non-adherence of the sample was 12.6% according to the mobile application, while 60% of caregivers self-scored as adherent according to the MAT. The only significant and positive correlation was between overall adherence and the caregiver’s education. The average degree of pruritus decreased over time and during treatment, independently to the level of m-health app usage. **Conclusions:** The adherence to chronic treatments may be improved through m-health apps, although further studies are needed to gauge their actual usefulness in supplementing known adherence determinants.

## 1. Introduction

The role of a pharmacist includes having knowledge of veterinary medicinal products in general, the validation of veterinary prescriptions, and the dispensing of such products with the same diligence and attention given to medicines for human use [1,2]. It is well known that adherence to therapy is crucial for the treatments’ success, including the control of the symptoms of chronic pathologies, usually requiring a tailored approach [3]. The usage of a mobile app that informs about medication use and reminds caregivers of medication schedules, both in humans and animals (such as pets), can be of great advantage to promote treatment adherence and optimal treatment outcomes [4,5].

Canine Atopic Dermatitis (CAD) is a chronic disease that is common in dogs and has no cure. Affecting 10%–15% of the general dog population and being chronic in most cases, similarly to atopic eczema in people, the condition has a great impact on the animals’ and their owners’ quality of life. The chronic and often severe course of the disease, the recurrent flares, and its lifelong management are challenging for both owners and veterinarians [6,7].

The currently available symptomatic therapies for CAD typically require a multimodal approach in which, usually, two or more expensive medications are to be given simultaneously, with varying frequencies of administration and dosages over time [8]. To be effective, the treatment must include a reduction of pruritus and inflammation, treatment and prevention of secondary infections (both from the skin and ear canals), allergen avoidance or desensitization, and an improved skin barrier function. As such, a single drug is insufficient to control the symptoms, and complex drug regimens are normally required. For these reasons, it has been proved that CAD management affects both dogs and their owners on a physical, emotional, and financial level [7]. The financial burden of some veterinary products, such as immunotherapy, Cytopoint^®^ or Atopica^®^, can easily exceed hundreds of Euros. Thus, initiatives such as mobile health (m-health) apps that are designed to improve, amongst other features, the correct use of complex and demanding therapies, may prove to be a good resource in aiding the management of chronic conditions, including those affecting pets. These apps look particularly promising concerning medication adherence, treatment effectiveness, and expenses control.

The purpose of this study was to evaluate the adherence of animal owners to therapy through m-health use. It aimed to contribute to the improvement of compliance with complex therapeutic regimens, achieving a faster and more effective control of CAD symptoms.

## 2. Materials and Methods 

The present pilot study followed an exploratory, descriptive, and cross-sectional design, using a convenience sampling to approach the potential of m-health in selected CAD cases [9]. One critical feature of such an application would be the back-office access for researchers to medication-related data, e.g., drug, dosage, and administration moment. Former students from the Faculty of Veterinary Medicine of the University of Lisbon (FVM-UL) have developed an m-health app for veterinary purposes—Petable^®^ (https://petable.care/). This app allows users to set medication alarms and to register the administration of medicines for therapy management and control. Through a partnership agreement, the researchers were granted access to the adherence data. After the prescription, the participant pet owners or caregivers registered both the medication and regime on the app. This was performed under the supervision of the veterinarian. The researchers had secure and anonymous web access to the application back-office, thus being able to confirm the medication that was registered as been administered.

The study population consisted of dogs and their owners, opportunistically and sequentially recruited at the dermatology service of the FVM-UL teaching hospital. From March to September 2019, the field researcher was present in a total of 70 CAD medical appointments, from which 42 dogs were selected according to the inclusion/exclusion criteria defined below:Owner or caregiver age over 18 years.Having a smartphone with an Android or IOS operating system.Being a new user of the Petable^®^ mobile application, agreeing to its active use.Owner or caregiver exclusively using the Petable^®^ application to this end.Dog with pre-diagnosis of CAD (1st-time diagnosis).Dog with established therapy for CAD.Dog with multiple medicines for CAD treatment.

The 42 cases were needed to reach the study sample of 30 cases, i.e., pet owners that agreed to participate after giving informed consent and correctly installing and using the Petable^®^ mobile application [10]. The sample size (N ≥ 30) was defined following statistical theory (the central limit theorem) for the assumption of a sample mean normal distribution [11], but without aiming to have sample power calculations or statistical representation. After the purchase of the prescribed medicinal products (most at the teaching hospital, some at community pharmacies), the pet owners agreed to use the m-health app in the following six weeks after the consultation. They were responsible for recording if the medication was (or was not) given or applied as directed to their pets, a single condition scoring 1 (or 0) for each medication, respectively.

The study instruments included, besides the mobile application, a formulary to register the owners’ background information (e.g., age, gender, education, number of pets) as well as the dogs’ clinical data, e.g., pruritus degree and the prescribed medication, obtained from clinical records. The pruritus scale varied from 0 to 10, where 0 corresponded to no itching and 10 was the maximum degree of itching, where the dog intensely rubs and scratches himself, day and night [12]. To measure treatment adherence, a published Portuguese tool (the *Medida de Adesão aos Tratamentos* or the MAT questionnaire) [13] was used as a concurrent medication adherence assessment. This tool (MATv) was composed of seven dichotomous items (“yes” = 0, “no” = 1) and was scored after summing the items, from 0 to 7. The instrument questions were adapted into a veterinary context by direct changes in the wording, e.g., “Have you ever forgotten (to take/) to give your pet the medication for (your/) his disease?”, without pursuing psychometric validation. The MATv was answered by all pet owners at the end of the study (week 6) by a telephone interview, and any pet owner with a score equal to or above the median value was considered adherent [13]. Treatment non-adherence assessed by the Petable^®^ app was calculated when there were more than 20% missing doses [14]. For data analysis, the owners were classified into two subgroups, being overall medication non-adherent (i.e., 0) all owners established as non-adherent through any of the measures, i.e., Petable® and/or MATv. Both groups were studied in relation to any statistically significant associations.

The data analysis was accomplished using the IBM^®^ SPSS^®^ (version 26) statistical program, consisting of, besides descriptive statistics, the Chi-square, Analysis of Variance (ANOVA), and Spearman’s tests to evaluate non-parametric correlations between continuous variables. Values of *p* < 0.05 were considered statistically significant.

The study received ethical clearance from the Institutional Review Board of the FMV-UL (Reference # 013/18).

## 3. Results

### 3.1. Study Sample

The study sample consisted of 30 dogs, in equal numbers for the animals’ sex. The animal sample ages ranged from 2 to 10 years old, with a mean age of 5.1 years (SD 1.97). Regarding the breeds, 16 different breeds were sampled, with 16.7% being Labradors (the most frequent breed), while 23.3% were of an undetermined breed.

Each animal had one owner who assumed the caregiver role, i.e., being responsible for administering the therapy. Of the 30 owners, 22 were female, aged between 18 and 79 years, with a mean age of 38.9 years (SD 16.01). The owners’ levels of education varied between 9th grade and master’s degree: 63.3% (19) had a university degree, while 36.7% (11) had completed 12th grade or attended high school (9th grade). Most households (36.7%) consisted of two people, followed by 33.3% of households consisting of four people. The average time for owning the pet was 3.6 years, with 40% (12) of owners being guardians of the animal for 5 or 6 years, and 16.7% (5) for 2 years. In total, 43.3% (13) of the owners had animals other than the dog with CAD under treatment. Most owners (46.7%) were simultaneously collecting medication for the treatment at the FMV-UL school hospital and a community pharmacy.

### 3.2. CAD Medication 

The treatment used in this sample involved a total of 30 different medicinal products, composed of medications with a single active substance to combine formulations and adjuvants (e.g., washing preparations) for internal or external use. These products included, most frequently, antibiotics (e.g., amoxicillin, gentamicin topical, marbofloxacin), antifungals (e.g., clotrimazole, climbazole, miconazole), corticoids (e.g., dexamethasone, betamethasone, prednisolone), as well as other substances such as immunosuppressants (e.g., cyclosporine), and agents specifically developed for CAD pruritus control (e.g., oclacitinib, anti-IL-31 antibodies).

During the six weeks of the study, 135 medications in total were prescribed by veterinarians to the 30 animals. The CAD treatment in this sample involved the prescription and use of at least two different medicines up to a maximum of seven different products per animal. Only two dogs were prescribed two products, while seven different products were also prescribed to another two. The mode was four different medicinal products (10 animals). There were 17 different medication regimes, varying from one administration daily, weekly, and monthly until the end of the package, to complex administration regimens such as one daily administration during the first seven days, followed by administration every other day for five days, and finishing with a half dose every other day for the last five days of treatment. The treatment duration also varied for the same medication: for instance, the Douxo Pyo^®^ could be used every other day for one week, daily for two weeks, once a week (or twice a week for two weeks), three times a week for one month, and once every two weeks. The duration varied frequently for the same medication, with a reevaluation of the CAD symptoms after two to four weeks of treatment.

Table 1 shows the prescribed products, their prescribed frequency (within the 30 animals), as well as the most frequent administration scheme, and the total number of different administration possibilities in the study sample.

The sample average of pruritus reported by the owners is presented in Figure 1. It was possible to confirm the overall reduction of the sample’s main CAD symptom, as expected. After starting the treatment, the reported pruritus decreased, with a significant difference when comparing the symptoms at the consultation and after six weeks of treatment.

### 3.3. Medication Adherence Data

Not all medications available to be purchased at the faculty’s teaching hospital were bought by the owners, indicating intentional non-adherence [3]. This was confirmed by a thorough analysis of the clinical records for each animal. It was observed that 10 prescribed medications were not purchased, which corresponds to 7.4% of primary non-adherence. The medicinal products concerned were several presentations of Cortavance^®^, Douxo^®^, Otodine^®^, Cytopoint^®^, and the ‘maintenance allergen-specific immunotherapy’ (monthly “allergy shots”).

At home, the Petable^®^ app indicated that the owners missed more than 20% of the doses of prescribed medications for nine products. At this point, only the medications actually purchased were monitored by the application, giving a 6.7% non-adherence rate. The products concerned were: Advantan^®^ (twice a day for 7 days), Canesten^®^ (every other day), Cortavance^®^, Douxo^®^ Pyo Shampoo and Mousse (for weekly and twice-weekly baths), Douxo^®^ Pyo Shampoo (for two baths per week), Douxo^®^ Pyo Mousse, Malaseb^®^ (for a monthly bath), and maintenance immunotherapy (a monthly vaccine).

The total non-adherence was estimated at 14.1%, i.e., from the 135 prescribed products, 19 were not purchased or administered by owners as expected. The products with the lowest adherence were Cortavance^®^ (SDS) and Douxo^®^ Pyo Shampoo (for weekly bathing). This occurred with 8 out of the 30 animals. Concerning MATv results, the average value was 6.27 (SD = 1.048). All owners admitted that they had not given the medication on time (Question 2), while Question 5 (“Have you ever given your pet more medication or a higher dose of medication than your veterinarian indicated on your initiative because the pet got worse?”) showed no variance in response (100% “no”). Of the 30 owners, 18 scored seven on the questionnaire, meaning that 40% of owners were non-adherent according to MATv. Only the owners whose score was equal to or higher than the median (in this case, 7) were considered adherent.

### 3.4. Adherence versus Animals’ and Owners’ Features

All animals’ and owners’ background variables were statistically studied as possibly associated with the overall medication non-adherence behavior. The statistical differences were assessed from all available variables and the only significant difference found (ANOVA F = 3.28, *p* = 0.027) was in the owners’ education: the overall mean adherence of the owners who attended high school (9th grade) (50%) was significantly lower compared to the owners from all other schooling groups (group average 92%). This result emerged mainly from Petable^®^ since no statistical differences were found in mean MATv scores between the different educational levels. Nevertheless, there was a significant non-parametric correlation (*p* < 0.01) between the overall adherence and the MATv score. Additionally, when the mean MATv score was tested for the subgroups classified as adherent and non-adherent, there was a statistically significant difference (t = −4.75, *p* < 0.05).

In terms of the pets’ pruritus over the six weeks, and comparing the animals receiving the treatment as directed with those not receiving, there were significant differences between the two groups in the reported pruritus at the third week of treatment (t = 2.07, *p* = 0.04), being almost significant by the sixth week (*p* = 0.06), i.e., dogs receiving treatment as prescribed improved sooner than those not receiving as directed. This effect was attenuated with time and the ongoing treatment.

## 4. Discussion

This study aimed to gauge the impact of using an m-health application on improving medication adherence when employing a demanding treatment in third party patients, i.e., caregiving to pets with CAD. There are reported initiatives in this subject area [15], and the topic has received the attention of researchers according to recent reviews [16,17]. However, according to the authors’ best knowledge, this is one of the first published studies with veterinary medication adherence in pet samples.

The gender and age of owners were not related to medication adherence in this sample. The published literature shows that women are, in general, more adherent than men [18,19], and that having a caregiver does not necessarily improve medication adherence [20]. Although the sample had a mean age below 40 years, the lack of association between adherence and owners’ age might suggest an opportunity to use m-health apps independently from the age of the caregiver.

On the contrary, the owners’ education level was shown to be important in the maintenance of the treatment regime according to the veterinarian directions. This is aligned with previous research, showing that schooling is positively associated with health literacy [21], which in turn explains the degree to which owners are able to know and manage the treatment medications [22,23]. While the household composition, the time since the guardian has the animal, and having more animals did not seem to influence medication adherence, other factors might have had influence, such as the features of the products. The relatively higher prices of hospital products (one package of Cortavance^®^ costs 21.50 € and one Douxo^®^ Pyo Shampoo 16.80 €) compared to most veterinary pharmaceuticals purchased in community pharmacies, the difficulties in administration (spraying in areas where dogs have inflammation and itching or using the shampoo correctly in the bath, allowing a contact time of 5 to 10 min), the particular dosage (e.g., weekly administration that may be easy to forget, weekly showers requiring time, availability and physical exertion), or even the poor prior experience with the product in symptoms’ control, all are determinants of treatment adherence [24].

As expected, there was a relationship between using the medication and the decreased pruritus over time for all dogs, although CAD symptom improvement between adherent and non-adherent to treatment was not possible to confirm. While an overall treatment success was observed, the complexity of the medication and regimes introduced too much variance to be detected within the present small-sized sample and study length. 

Adherence to therapy measured by Petable^®^ presented an important higher percentage than adherence measured by the MATv questionnaire. This might indicate better control in medication usage than what is perceived by owners. When using Petable^®^, the owners should mark how each treatment was done at the time of administration, which could have been neglected knowing the app does not have a feedback mechanism in case of missing doses (a feature to be implemented). MATv was much more sensitive to the owners’ memory biases, adding to the overarching questions and dichotomous answers, which do not capture individual issues when administering treatments. For example, a guardian who replied that he or she had forgotten to administer one medication (“yes” answer to Question 1) was no longer considered to be adherent in our study. The questionnaire did not detect which medication was not administered and how many medications were not correctly used. Additionally, the MATv might have been prone to a contamination effect from the usual behaviors when the owners use human medications. On the other hand, the MATv questionnaire allowed us to see if owners have ever missed directions, providing reasons and explanations that were missing in Petable^®^. 

Several study limitations restrict the opportunity to conclude about the potential usefulness of the present m-health app for increasing medication adherence in CAD. Firstly, a reduced sample size might have affected the surfacing of stronger differences and associations. Secondly, adherence was measured overall with no control for overuse or by therapeutic groups, adding to the acquisition of products made in other outlets that were not registered. Although it is based on a well-known instrument with a Portuguese validated version, the MATv did not undergo any further validation of psychometry and other proprieties. A satisfaction survey for using the app was not achieved, limiting the conclusions of the m-health app’s usefulness.

## 5. Conclusions

As in many areas of present human lives, m-health apps are contributing to the better management of one’s health, including the improvement of coping with medication directions for patients. This study stressed that non-adherence to treatments is associated with less education, and may be reduced through additional resources such as m-health apps. Further studies are needed to gauge the actual usefulness of m-health apps in the supplementing of known adherence determinants and the control over chronic conditions. 

## Figures and Tables

**Figure 1 pharmacy-08-00038-f001:**
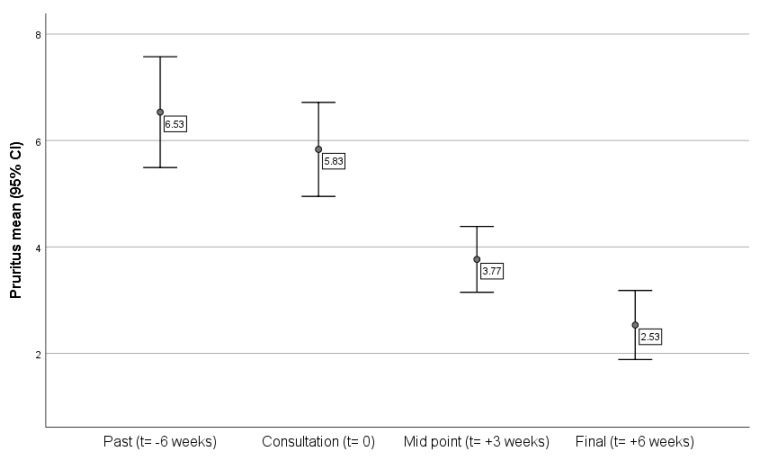
Sample’s pruritus mean before and during the study period.

**Table 1 pharmacy-08-00038-t001:** Prescribed medications organized by the descending frequency of prescription and administration regimes (N = 30).

Active Substances [Brand Name] EXTernal/INTernal Use {Cost} ^1^	Prescription % (N = 30)	Most Frequent Regime (N = 30)	Nº of Different Regimes (Max. 17)
Chlorhexidine + Climbazole [Douxo Pyo Shampoo, Spray, Mousse, Cleaning Disks^®^] EXT {$$}	96.7% (20)	Once a week (11)	7
Hydrocortisone Aceponate [Cortavance^®^] EXT {$$}	66.7% (29)	Once a week (8)	7
Prednisolone [Lepicortinolo^®^ or Prednicortone^®^] INT {$}	36.7% (11)	Once a day (8)	3
Oclacitinib [Apoquel^®^ 5,4 mg e 16 mg] INT {$$-$$$}	33.3% (10)	Once (4) or twice a day (4)	4
Lokevitmab [Cytopoint^®^] INT {$$-$$$}	30.0% (9)	Once a month (9)	1
Cyclosporine (oral solution) [Atopica^®^ e Cyclavance^®^] INT {$$$}	23.3% (7)	Once a day (7)	1
Methylprednisolone Aceponate [Advantan^®^] EXT	20.0% (6)	Twice a day (4)	4
Immunotherapy (allergen-specific) {initial $$$, maintenance $$$}	13.3% (4)	Once a month (3)	2
Itraconazole [Sporanox^®^ or Itrafungol^®^] INT	13.3% (4)	Once a day 1st week, once at weekends for a month (3)	2
Miconazole + Gentamicin + Hydrocortisone [Easotic^®^] EXT	10.0% (3)	Once a day (2)	2
Miconazole + Prednisolone + Polymyxin B [Conofite^®^] EXT	10.0% (3)	Twice a day (3)	1
Otodine^®^ + Dexamethasone [Vetacort^®^] EXT	6.7% (2)	Once a day (1)	2
Chlorhexidine + Miconazole [Malaseb^®^] EXT	6.7% (2)	Twice a week (1)	2
Clotrimazole [Canesten^®^] EXT	6.7% (2)	Every other day (1)	2
Cyclosporin (eye drops) EXT	6.7% (2)	Once a day (1)	2
Ketotifen [Zaditen^®^] EXT	6.7% (2)	Twice a day (2)	1
Allopurinol [Zyloric^®^] INT	3.3% (1)	Once a day (1)	1
Amoxicillin + Clavulanic Acid [Clavamox DT^®^] INT	3.3% (1)	Twice a day (1)	1
Cefalexin [Tsefalen^®^] INT	3.3% (1)	Twice a day (1)	1
Chlorhexidine [Otodine^®^] EXT	3.3% (1)	Once a week (1)	1
Clotrimazole (eye drops) [Diomicete^®^] EXT	3.3% (1)	Twice a day (1)	1
Dexamethasone Phosphate [Dexafree^®^] EXT	3.3% (1)	Once a week (1)	1
Fusidic Acid + Betamethasone [Isaderm^®^] EXT	3.3% (1)	Twice a day (1)	1
Isotretinoin [Roacutoan^®^] INT	3.3% (1)	Twice a day (1)	1
Marbofloxacin [Marbocyl^®^] INT	3.3% (1)	Once a day (1)	1
Neomycin + Triamcinolone + Nystatin + Thiostrepton [Panalog^®^] EXT	3.3% (1)	Once a day (1)	1
Pentoxifylline [Trental^®^] INT	3.3% (1)	Twice a day (1)	1
Prednisolone (eye drops) EXT	3.3% (1)	Twice a day (1)	1
Squalene + Chamomile + Salicylic e Tannic Acids [Otoact^®^] EXT	3.3% (1)	Three times a week (1)	1

^1^ Cost per usual package of the products available at the FMV-UL school hospital: $ below 10 €, $$ between 10 € and 50 €, $$$ above 50 € (prices may vary with dosage). Other medications can be purchased in community pharmacies, with varying price ranges.

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
