# Peer review of "Exploring Medication Adherence Using M-Health: A Study from Veterinary Medicine"

_pharmacy, 2020, doi:10.3390/pharmacy8010038_

Round 1
Reviewer 1 Report
This is a pilot study of the effect of a health application that alerts administration time on adherence for 30 canines with chronic canine atopic dermatitis and their caregivers. The following comments are provided in order to strengthen the manuscript.
Title is descriptive and accurately summarizes the study.
Abstract - Numbers in parantheses can be removed. The conclusion should state what further developments are needed to improve adherence.
Key words - there is a MeSH term for adherence, canine atopic dermatitis, and pruritis, but not for m-health or pet owners. I would suggest 'technology assessment, biomedical'; 'drugs, veterinary'; and 'pets' and eliminate m-health and pet owners because these have MeSH terms and will improve the manuscript's searchability in pubmed.
Methods - please indicate how this m-health application, Petable, is selected. If there are no others on the market, please indicate this. Sample size is good for a pilot study. Statistics are appropriate for analysis.
Table 1 - there is no noticeable order for the table's line items. Since frequency of administration is known to have a positive correlation with adherence, consider listing the items by frequency of administration. Mousse is misspelled in the first listing (Mouse). Not all listings have an associated cost to the caregiver. Cost could have a negative effect on adherence if the caregiver could not afford the prescription. The authors could make this more pronounced in the results.
Discussion - the authors indicate that apps have been tested, but in other populations (i.e. human therapies). However, if there is no published literature regarding the impact in animals, the authors could accentuate this point.
Conclusion - instead of schooling, I would suggest level of education. The last sentence could be imported to the abstract, since it is a clearer description.
References - slight changes for mdpi style are needed.
Author Response
Reviewer 1 comments
The authors would like to thank the reviewer for the comments received. Please find our responses below each comment. Thank you.
Abstract - Numbers in parantheses can be removed. The conclusion should state what further developments are needed to improve adherence.
R: Numbers removed from the abstract. Conclusion rewriting to accommodate the reviewer's suggestion. Now reads as “Chronic treatments’ non-adherence may be reduced through m-health, although further studies are needed to gauge the actual supplementing of usual caregiver-centered determinants.”
Key words - there is a MeSH term for adherence, canine atopic dermatitis, and pruritis, but not for m-health or pet owners. I would suggest 'technology assessment, biomedical'; 'drugs, veterinary'; and 'pets' and eliminate m-health and pet owners because these have MeSH terms and will improve the manuscript's searchability in pubmed.
R: Suggestions accepted. The keywords m-health and pet owners were deleted. The MeSH terms added were “veterinary pharmaceuticals” MeSH Unique ID: D019155 and “mobile health” MeSH Unique ID: D017216. Thank you.
Methods - please indicate how this m-health application, Petable, is selected. If there are no others on the market, please indicate this. Sample size is good for a pilot study. Statistics are appropriate for analysis.
R: The passage now read as: “One critical feature of such an application would be researchers’ back office access to medication-related data e.g. drug, dosage, and administration moment. Former students from the Faculty of Veterinary Medicine of the University of Lisbon (FVM-UL) have developed an m-health app for veterinary purposes – Petable® (https://petable.care/). This app allows to set medication alarms and to register medicines administration for therapy management and control. Through a partnership agreement the researchers were granted access to study adherence data.”
Table 1 - there is no noticeable order for the table's line items. Since frequency of administration is known to have a positive correlation with adherence, consider listing the items by frequency of administration. Mousse is misspelled in the first listing (Mouse). Not all listings have an associated cost to the caregiver. Cost could have a negative effect on adherence if the caregiver could not afford the prescription. The authors could make this more pronounced in the results.
R: Table 1 was organized by descending frequency of prescription. To make this clear, the Table title now reads as “Prescribed medications organized by descending frequency of prescription and administration regimes (N=30)”. Word Mousse is now corrected. In relation to costs, for the products purchased at the school hospital, the price was known, while the other products were bought in community pharmacies with varying costs. This information is now stressed in Table 1 footnote, as well as in the Discussion: “The relatively higher prices of hospital products (one package of Cortavance® costs 21.50€ and one Douxo® Pyo Shampoo 16.80€) compared to most veterinary pharmaceuticals purchase in community pharmacies, …”. Thank you.
Discussion - the authors indicate that apps have been tested, but in other populations (i.e. human therapies). However, if there is no published literature regarding the impact in animals, the authors could accentuate this point.
R: The following sentence was added in Discussion: “However, according to the authors’ best knowledge, this is one of the first published studies with veterinary medication adherence in pet samples.” Thank you.
Conclusion - instead of schooling, I would suggest level of education. The last sentence could be imported to the abstract, since it is a clearer description.
R: Changes accepted. The abstract now readers in the conclusion “Chronic treatments’ adherence may be improved through m-health apps, although further studies are needed to gauge the actual supplementing of known adherence determinants.”
References - slight changes for mdpi style are needed.
R: These were corrected. Thank you.
Reviewer 2 Report
1) The study should be reported using ad hoc guidelines and checklists, such as the STROBE.
2) Please report explicitly the power sample analysis, without referring only to reference 9.
3) Provide a flowchart with incluson/exclusion critera of the sample used in the study.
4) Please report the psychometric properties of the instrument.
5) Axis y label of figure 1 should be mean.
Author Response
Reviewer 2 comments
The authors would like to thank the reviewer for the comments received. Please find our responses below each comment. Thank you.
1) The study should be reported using ad hoc guidelines and checklists, such as the STROBE.
The STROBE (Strengthening The Reporting of OBservational Studies in Epidemiology) checklist follows an epidemiological perspective. The present pilot study does not have such pretention, being exploratory by nature. Nevertheless, and following the reviewer’s suggestion, the checklist proposed by the MInCir Initiative, published in the International Journal of Morphology, 35(1):72-76, 2017 was followed. Additionally, in the Methods section, the opening sentence was rewritten to strength the empirical nature of the present study and the previous reference added: “The present pilot study followed an exploratory, descriptive and cross-sectional design, using a convenience sampling to approach the potential of m-health in selected CAD cases[9].”
2) Please report explicitly the power sample analysis, without referring only to reference 9.
The authors understand the reviewer's concern with epidemiological features. Following on the authors’ preceding reply and as stated in the manuscript Methods as well as in the Study Limitations, a convenient sample was used, with no attempt to reach sample power for statistical representation. The following sentence was introduced: “The sample size (n>=30) was defined following statistical theory (the central limit theorem) for the assumption of a sample mean normal distribution,[11] but without aiming to have sample power calculations or statistical representation.”
3) Provide a flowchart with incluson/exclusion critera of the sample used in the study.
The sampling procedures included 2 main steps, following an opportunistic sampling strategy. The 1st step comprised the sequential screening of all dogs (70 during the study period) with an appointment for a CAD consultation, using the inclusion/exclusion criteria. This step selected 42 dogs. The 2nd step comprised the selection of 30 sample cases i.e. those pet owners that agree to participate after informed consent, installed and used the mHealth app. If the reviewer considers that a 2 steps sampling strategy requires a flow chart, authors will gladly provide the additional figure. Anyway, the passages read now as: “From March to September 2019, the field researcher was present in a total of 70 CAD medical appointments, from which 42 dogs were selected according to the inclusion/exclusion criteria defined below:…” and “The 42 cases were needed to reach the study sample of 30 cases, i.e. pet owners that agreed to participate after informed consent and correctly installed and used the Petable® mobile application[10].”
It should be noted the study did not start from a population-base, with case selection by applying inclusion/exclusion criteria. From the dogs stepping into a specific CAD consultation (70 animals), cases were opportunistically recruited by order of appearance, according to the inclusion/exclusion criteria (42 animals). Sample cases (up to 30) entered the study if agreeing to participate after informed consent and installing the mHealth app.
4) Please report the psychometric properties of the instrument.
Reporting main properties such as construct and content validity measures, as well as reliability values, would deliver another study in the authors’ opinion.
The instrument was modified to fit the veterinary context, with direct changes in wording, e.g. “Have you ever forgot (to take)/to give your pet the medication for (your)/his disease?”. Questions were dichotomized (0,1) and could have worked as independent items. It is stated in Methods, as well as in Study Limitations that no psychometric evaluations were aimed. Additionally, to guarantee pet owners’ full participation at the end of the study, the questionnaire was administered by a telephone interview, information now disclosed in the manuscript.
If the reviewer believes that psychometric properties are a critical issue, authors can make use of the first single question, described above. This approach follows a recent example published in the Journal of Medical Internet Research 2019;21(2):e13125. In this study, adherence was initially measured with one instrument (MMAS-8) but later reported as a single item measure from another instrument (MOS), with journal editors’ support. It was accepted the outcomes from the two measures were similar and the change would not substantially impact the interpretation of the findings and conclusions. The MAT Portuguese scale, also developed from Morisky et al. (1986), might be replaced by one single question, although losing some information richness. Looking forward to hearing from you. Thank you.
5) Axis y label of figure 1 should be mean.
- Axis y label corrected. Thank you.
Round 2
Reviewer 2 Report
The manuscript has been improved and can be accepted for publication.